# A compact design of MIMO patch antenna with high gain and symmetrical radiation pattern

**Phuong Kim-Thi**[1]*, **Thang Nguyen-Van**[2], **Dat Nguyen-Tien**[3], **Tung Bui-Thanh**[2]

**1** Faculty of Electrical and Electronics Engineering, Thuyloi University, Hanoi, Vietnam, **2** Faculty of Electronics and Telecommunications, VNU - University of Engineering and Technology, Hanoi, Vietnam, **3** Faculty of Electrical and Electronic Engineering, PHENIKAA University, Yen Nghia, Ha Dong, Hanoi, Vietnam

* phuongkt@tlu.edu.vn

**Data availability statement:** All relevant data are within the paper.

## Abstract

This paper presents a compact design of multiple-input multiple-output (MIMO) microstrip patch antenna with high gain and symmetrical radiation pattern. The proposed antenna consists of dual-polarized crossed patches and T-junction power dividers. This combination makes the proposed design achieve high gain and high isolation with compact size and not requiring any additional decoupling structure. Especially, the use of dual-polarized patch can help to decrease the number of radiating elements while exhibiting high gain radiation. Besides, symmetrical radiation pattern in the broadside direction is also obtained. A prototype of 2-port MIMO antenna with overall dimensions of $0.72\lambda \times 0.48\lambda \times 0.04\lambda$ is fabricated and tested. The measured operating bandwidth is from 4.74 to 4.87 GHz. Across this band, high isolation of better than 20 dB can be obtained with small element spacing of $0.005\lambda$. Besides, the measured far-field results observe symmetrical measured radiation pattern and broadside gain of 7.3 dBi. Further investigation also indicates that the proposed approach can be applied with large scale MIMO designs of 4 or 6 ports.

## Introduction

Microstrip patch antenna has been widely used in the multiple-input-multiple-output (MIMO) systems, which require unidirectional beam with compact and planar structures. When two or more identical patches are closely positioned, the strong mutual coupling between the adjacent elements will significantly deteriorate the system performance. Additionally, as there always exists a strong demand for device miniaturization, a compact MIMO antenna with simple structure and high isolation is preferred.

A huge of efforts have been implemented to suppress the mutual coupling between the MIMO elements by adding various types of decoupling networks. They could be band-stop structures [1–5] to block the coupling fields from the excited element to the others or additional coupling path, which are parasitic elements or neutralization lines [6–10] or, to minimize the effects of the original coupling path. Alternatively, metasurface and near-field

**Funding:** The author(s) received no specific funding for this work.

**Competing interests:** The authors have declared that no competing interests exist.

resonators [11–15] are also an effective solution in mutual coupling reduction when they are placed above the radiating elements. Self-decoupling is a promising method to decouple MIMO antenna with simple and compact layout [16–20]. Although they can obtain satisfactory isolation improvement, the designs with decoupling networks [1–8,11–14,16–18] inevitably result in more complex and bulky antenna structures. Meanwhile, a compact MIMO with small element spacing exhibits low gain radiation.

Several methods have been proposed to improve the gain of MIMO antennas. In [21–26], different types of superstrates such as dielectric slabs, frequency selective surface (FSS) are employed. These structures are generally positioned at a proper distance from the radiator, which significantly enhances the antenna size in the vertical direction. To the best of authors knowledge, there are few planar high-gain MIMO antennas reported in the literature [27–30]. Noted that all of them utilize a similar approach of combining two single-polarized radiating elements and one T-junction divider as one port of the MIMO system. It means that two-port MIMO antennas require four radiating elements. Obviously, the gain improvement is made at the expense of number of radiating elements, overall size, as well as element spacing. Furthermore, this approach also results in the asymmetrical radiation pattern in the coupling plane due to the unwanted radiation from the coupled elements.

In this paper, a different approach to design planar MIMO antenna with high gain radiation. The proposed design employs two dual-polarized crossed patches, in which a single crossed patch has two orthogonal excitation feeds, and two T-junction dividers. Consequently, high gain radiation is achieved with two radiating elements. Besides, due to the orthogonal arrangement, high isolation without requiring any additional decoupling structure is yielded. This configuration allows the MIMO system to radiate symmetrically in the broadside direction. These features distinguish the proposed design from the conventional approaches reported in [27–30]. It is also worth noting that compare to the related MIMO using T-divider, the proposed approach has the most compact size in terms of overall dimensions and element spacing as well. To make the design approach straightforward, the gain improvement is first discussed. Then, a dual-polarized element and its MIMO configuration are presented. Finally, measurement validation is implemented.

## High gain MIMO antenna solutions

Fig 1 shows different approaches to design high gain MIMO patch antenna. The first technique has a superstrate placed at a distance from the radiating elements. Depending on the reflection phase characteristic of the superstrate, the distance could be from quarter- to half-wavelength. Besides, large ground planes are also required to form a resonant cavity. Obviously, this approach inevitably increases the antenna size in both vertical and horizontal directions. The second approach utilizes multiple single-polarized patches and T-dividers. Here, one MIMO port is a combination of two radiating elements and one T-divider. Therefore, at least four radiating elements are necessary for a 2-port MIMO antenna. This method also has a trade-off with antenna dimensions. Additionally, these MIMO configurations always suffer from the asymmetric radiation problem in the coupling plane, which is caused by the disturbance from the coupled adjacent elements. To overcome these deficiencies, a proposed solution is based on the dual-polarized antenna and the T-divider. It can be seen obviously that this configuration simultaneously excites two similar polarizations on two patches. As a result, high gain radiation is achieved with only two radiating elements, rather than four elements. Meanwhile, the coupled components are arranged orthogonally to the excited ones, leading to high isolation without requiring any decoupling network. In terms of radiation patterns, the co-polarization components are excited simultaneously with equal

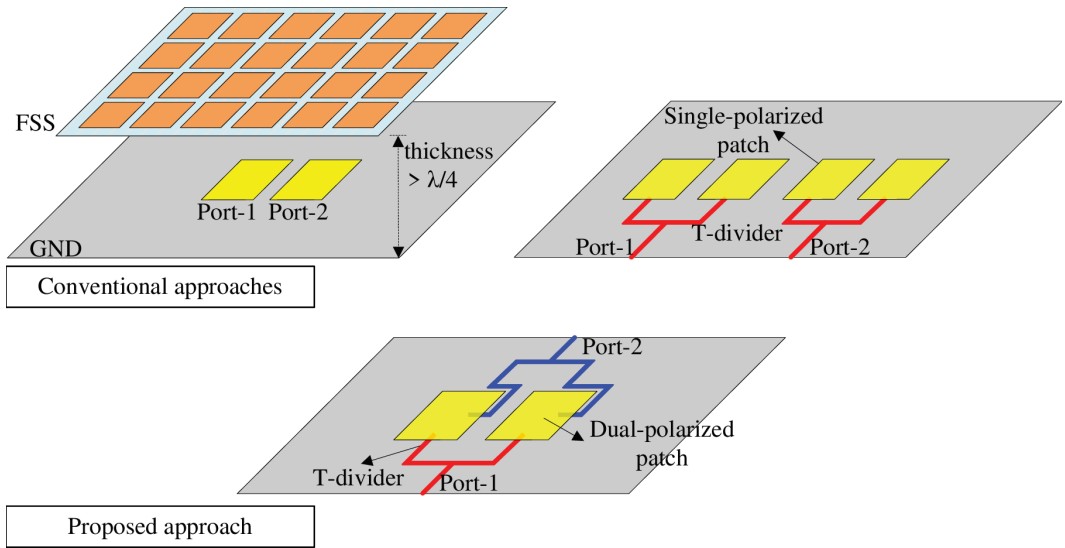

**Fig 1. Different solutions to design high gain MIMO antennas: antenna with FSS, single-polarized patch with T-divider, and dual-polarized patch with T-divider.**

phase and magnitudes. Therefore, symmetrical radiation pattern in the broadside direction can be obtained. Meanwhile, as the coupled parts are orthogonal to the excited parts, the radiation from them only affects the cross-polarization.

## Design a dual-polarized antenna

According to the discussion in the previous section, dual-feed dual-polarized is first considered. Fig 2 shows the geometry of the proposed crossed patch antenna with dual feeding positions. The antenna is printed on a top layer of a single Taconic RF-35 substrate with dielectric constant of 3.5 and loss tangent of 0.002. Two different coaxial cables are employed to excite the crossed patch at two feeding points, Port-1 and Port-2. The optimal design parameters are as follows: $W_s = 30$, $h_1 = 1.52$, $l = 18$, $w = 6$, $g = 2$ (unit: mm).

The simulated performance in terms of reflection and transmission coefficients ($|S_{11}|$, $|S_{21}|$), as well as radiation pattern at 4.8 GHz is presented in Fig 3. As observed, the antenna well operates at 4.8 GHz with $|S_{11}|$ less than –10 dB. Across this band, very high isolation of more than 40 dB is obtained. This is due to the position of the coupled port (Port-2), which lies on the null locus when the antenna is excited at Port-1. For demonstration, Fig 4 depicts the simulated electric field (E-field) distribution at 4.8 GHz on the crossed patch with Port-1 excitation. It is obvious that the null locus appears at the center, which makes the cross coupling between the two ports be suppressed. With respect to gain radiation, the antenna exhibits symmetrical radiation pattern around the broadside direction with a maximum value of about 6.0 dBi. Besides, due to the compact overall dimensions of $0.48\lambda \times 0.48\lambda \times 0.02\lambda$, the back radiation is quite high of about –9.0 dBi.

## MIMO configurations

### 2-port MIMO

The geometry in terms of top- and side-view of the proposed 2-port MIMO antenna is shown in Fig 5. For MIMO configuration, two crossed patches have four different feeding locations,

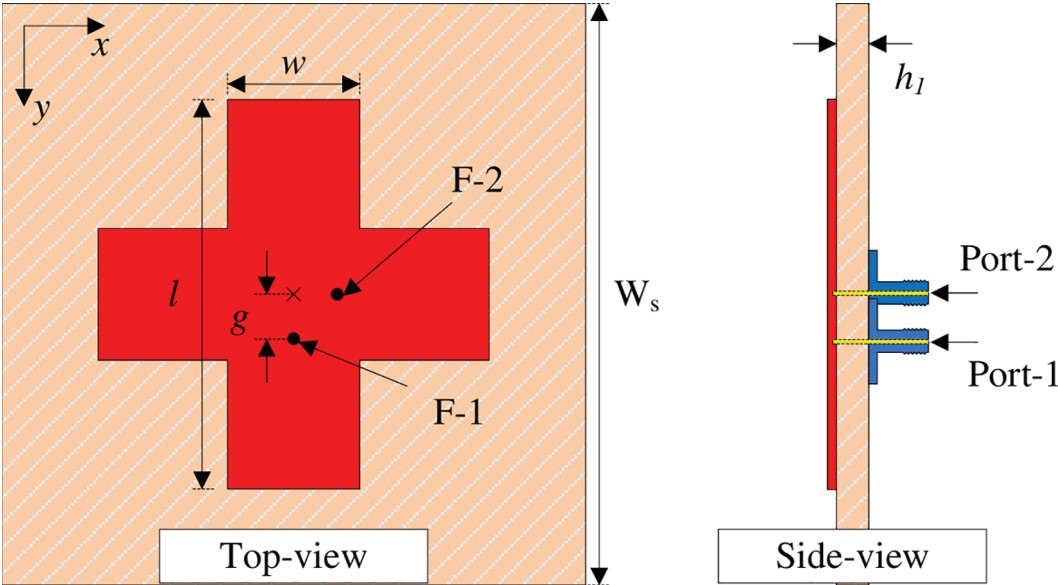

**Fig 2. Geometry of the proposed dual-polarized crossed patch antenna.**

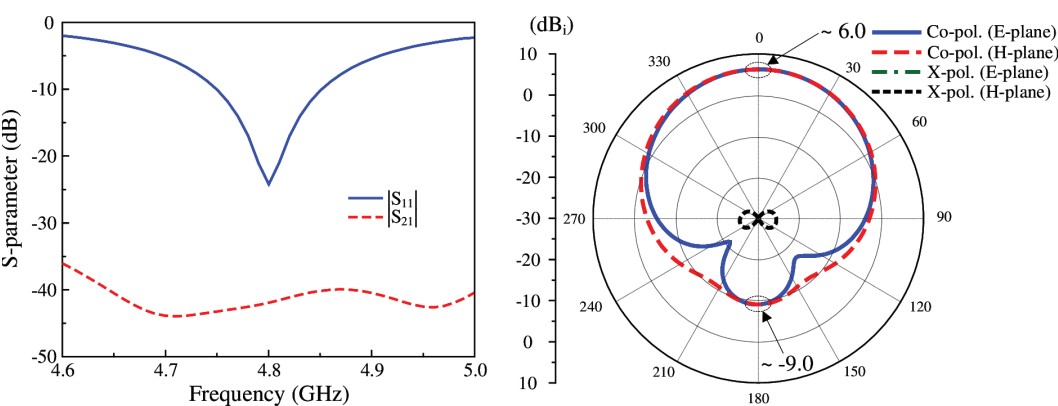

**Fig 3. Simulated S-parameter and radiation pattern of the crossed patch antenna.**

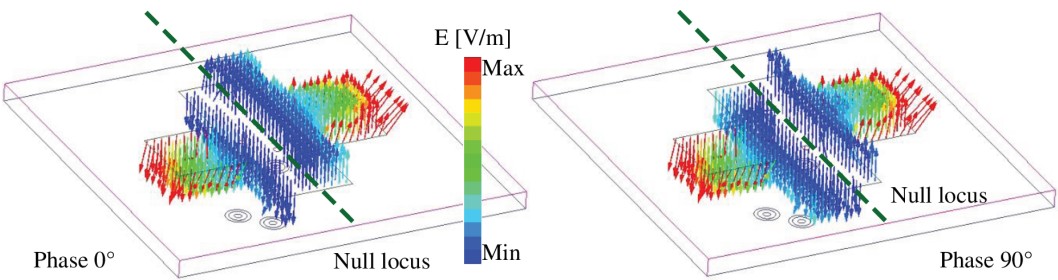

**Fig 4. Simulated E-field distribution at 4.8 GHz with Port-1 excitation.**

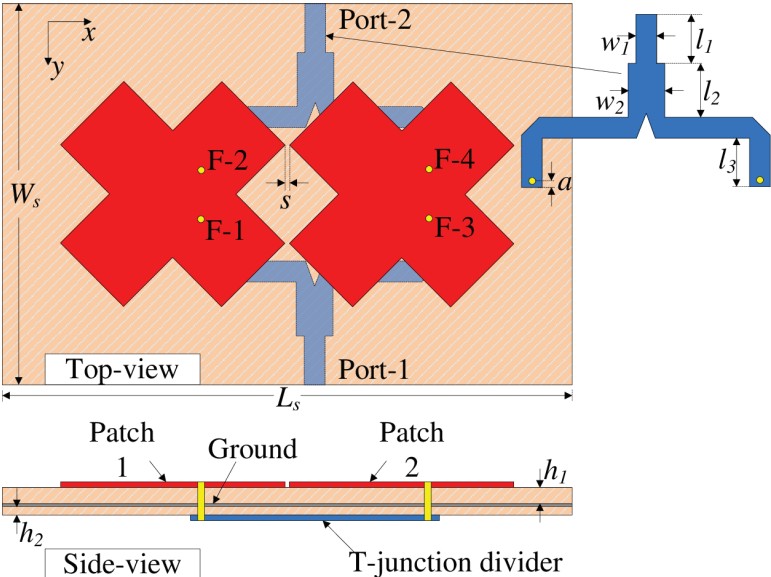

**Fig 5. Geometry of the proposed 2-port MIMO antenna.**

F-1, -2, -3, and -4. Two T-junction power dividers are employed to excite these patches. The T-divider has its input connected to either Port-1 or Port-2, while its outputs are connected to either (F-1, F-3) or (F-2, F-4). The principle for high isolation of the proposed design is based on two orthogonal linear polarizations. As seen in Fig 5, F-1 and F-3 will be excited for Port-1 operation. The coupling will occur between these feeding positions to the others, F-2 and F-4. However, the positions of F-2 and F-4 are orthogonal to F-1 and F-3. Consequently, high isolation will be obtained. To avoid disturbance to the antenna performance, the feeding network is printed on another Taconic RF-35 substrate and beneath the ground plane. The optimized design parameters are summarized in Table 1.

To design the 2-port MIMO antenna, two different arrangements of the crossed patches are considered. In the first case, these patches are positioned along x- and y-direction,

**Table 1. Optimized design parameters of the 2-port MIMO antenna.**

| Design Parameter | Value (unit:mm) |
|---|---|
| $L_s$ | 45 |
| $W_s$ | 30 |
| $h_1$ | 1.52 |
| $h_2$ | 0.76 |
| $l$ | 18 |
| $w$ | 7 |
| $g$ | 3 |
| $l_1$ | 4 |
| $w_1$ | 1.7 |
| $l_2$ | 4.2 |
| $w_2$ | 2.9 |
| $l_3$ | 3.8 |
| $a$ | 0.8 |
| $s$ | 0.3 |

designated as Ant-1 in Fig 6. Meanwhile, the other case has the patches rotated at 45°, designated as Ant-2 (shown in Fig 7). Noted that the MIMO antennas in both cases are optimized with similar overall dimensions for fair comparison. The simulated results of these antennas are presented in Figs 6 and 7. For the first case like illustrated in Fig 6, the simulated reflection coefficients with Port-1 and Port-2 excitations are not similar, leading to a small common bandwidth. Similar phenomenon can be observed in the broadside profiles. This is due to the different arrangements, in which the radiating parts with Port-1 excitation are positioned in H-coupled configuration, while the others with Port-2 excitation are arranged in E-plane configuration. In contrast, by rotating the patches at 45°, as observed in Fig 7, the simulated reflection coefficients for both ports are identical, which are less than –10 dB in the frequency range from 4.72 to 4.86 GHz. Across this band, the isolation is always better than 22 dB. With respect to broadside gain, the antenna can achieve a similar gain profile

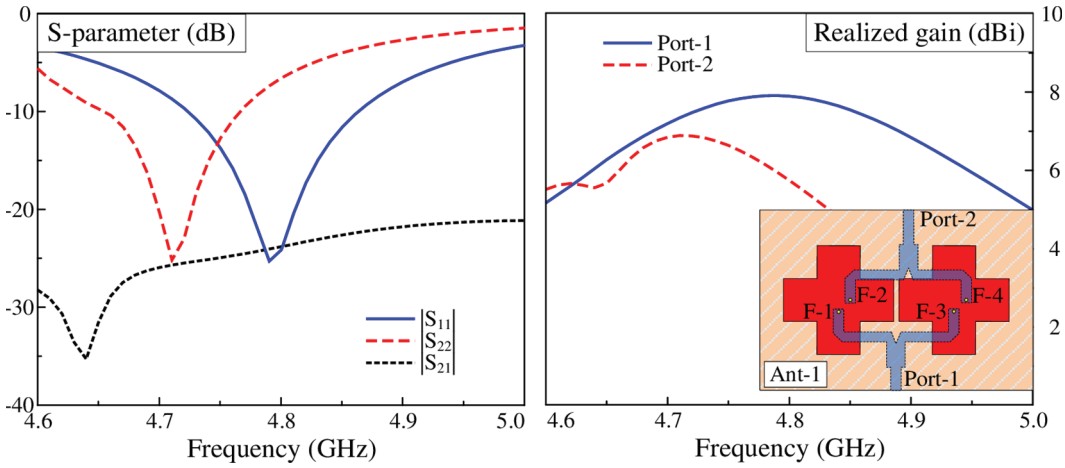

**Fig 6. Simulated performance of the first MIMO configuration - Ant-1.**

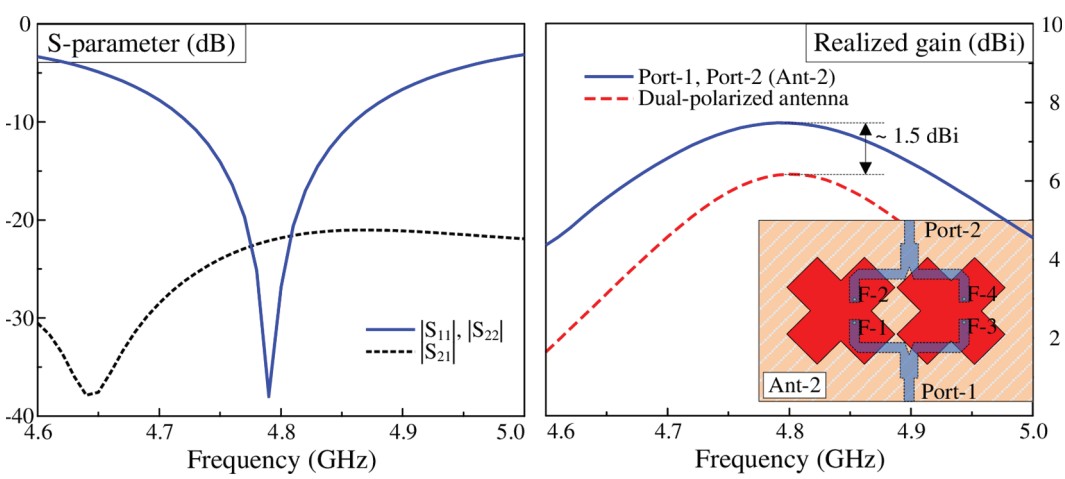

**Fig 7. Simulated performance of the second MIMO configuration - Ant-2.**

for both ports with a high gain of about 7.5 dBi. Noted that this value is about 1.5 dBi higher than the dual-polarized design discussed in the section about designing the dual-polarized antenna.

According to the simulated data of Ant-2, it also explains the advantages of using the dual-polarized crossed patch for MIMO configuration, rather than using the conventional dual-polarized square patch. For demonstration, the performance of the MIMO antenna using the square patch is presented in Figs 8 and 9. As discussed, the antenna should be rotated at 45° to ensure that the radiation characteristics of both ports are identical (as Ant-2). This antenna is optimized with similar overall dimensions as Ant-2. As observed, the matching and isolation characteristics of both antennas are similar. Using the square patch has the

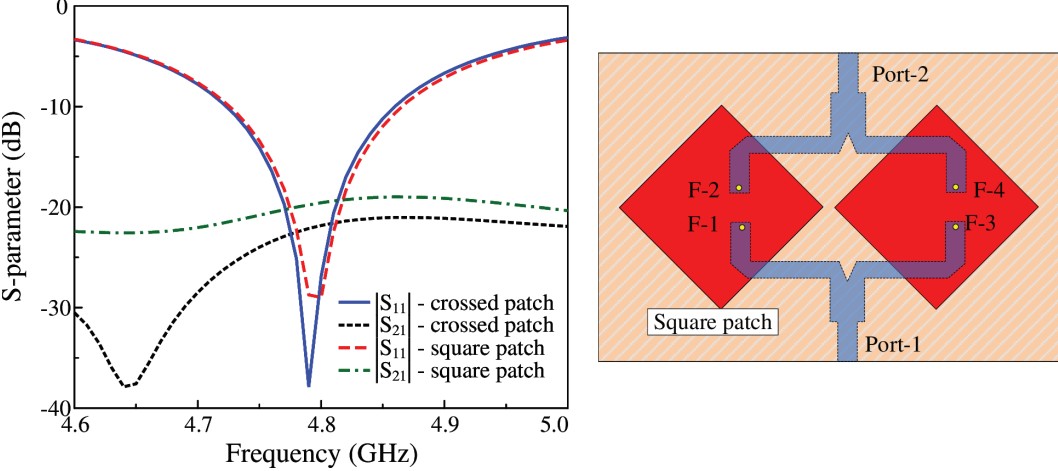

**Fig 8. Comparison between the square patch and crossed patch MIMO in terms of S-parameter.**

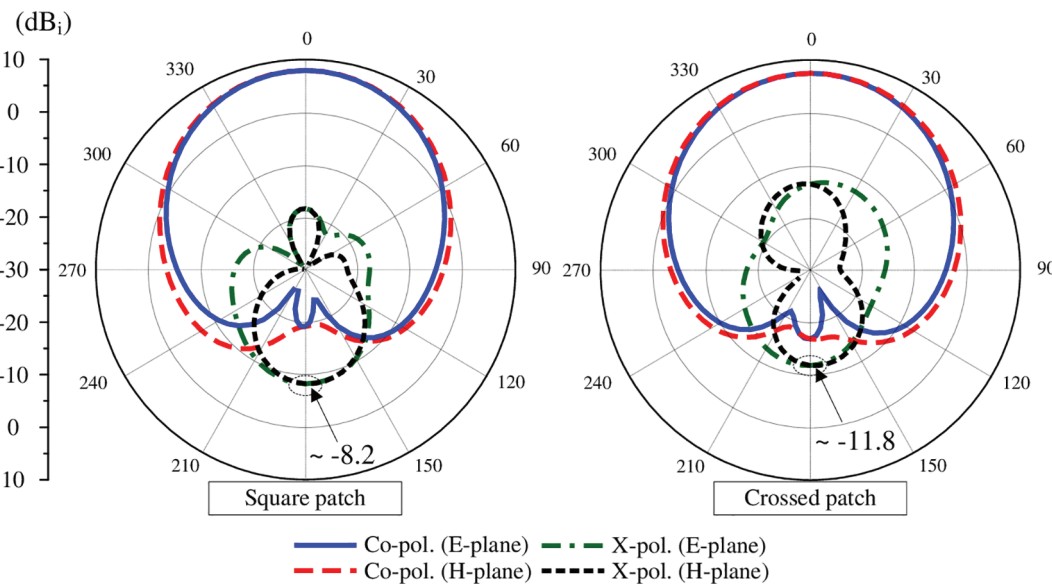

**Fig 9. Comparison between the square patch and crossed patch MIMO in terms of gain radiation patterns at 4.8 GHz.**

broadside gain of 7.9 dB. However, this antenna suffers from the high back radiation of about -8.2 dB. It is higher than that of the antenna using crossed patch, whose back radiation is about -11.8 dB. Meanwhile, the broadside gain of this antenna is about 7.6 dB. The reason behind is the large element spacing required for patch allocation, which makes the square patch close to the edge of the substrate and high diffraction accordingly. It is also worth noting that to achieve a similar performance in terms of S-parameter and broadside gain, the element spacing when using the square patch is 24 mm, which is significantly larger than the use of crossed patch with element spacing of 18 mm. This drawback will be more serious when the MIMO antenna is scaled up to multiple ports. In this case, requiring large element spacing will considerably increase the overall size of the antenna system.

Finally, the MIMO diversity features in terms of Envelop Coefficient Correlation (ECC) and Diversity Gain (DG) of the 2-port MIMO antenna are calculated and depicted in Fig 10. In the operating bandwidth (4.72–4.86 GHz), the ECC is much less than the acceptable value of 0.5. Meanwhile, the DG is almost close to the maximum value of 10 dB.

## Multi-port MIMO

The capability to scale up the proposed 2-port MIMO antenna to multi-port MIMO is investigated. Fig 11 shows two different configurations of 4-port and 6-port MIMO. The edge-to-edge spacing is similar for both antennas. The simulated S-parameter and far-field results of these designs are presented in Figs 12, 13 and 14. It is obvious that the impedance matching bandwidths for all cases are less than –10 dB around 4.8 GHz and the inter-port isolations are always better than 20 dB. Meanwhile, the radiation patterns of these MIMO antennas are strongest and symmetric around the broadside direction. Based on these results, it is obvious that the proposed approach to design compact high gain MIMO antenna can be scaled up to multiple-port MIMO arrays.

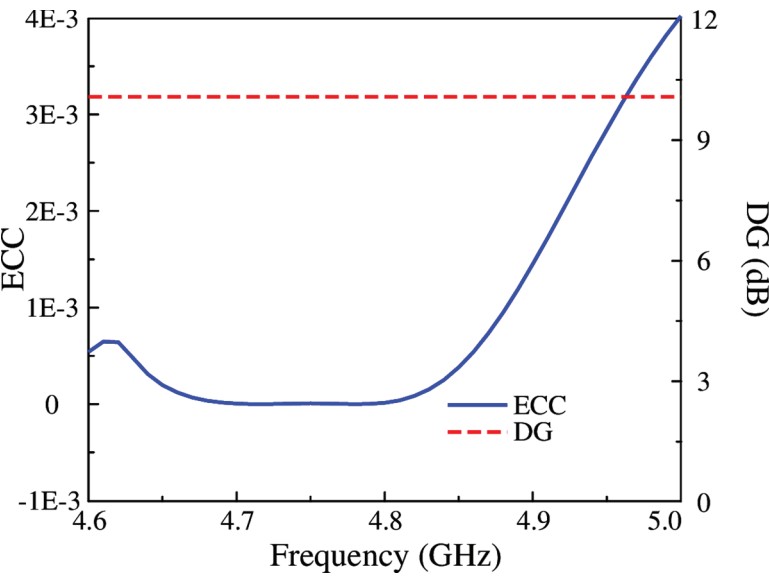

**Fig 10. Calculated ECC and DG of the proposed 2-port MIMO antenna.**

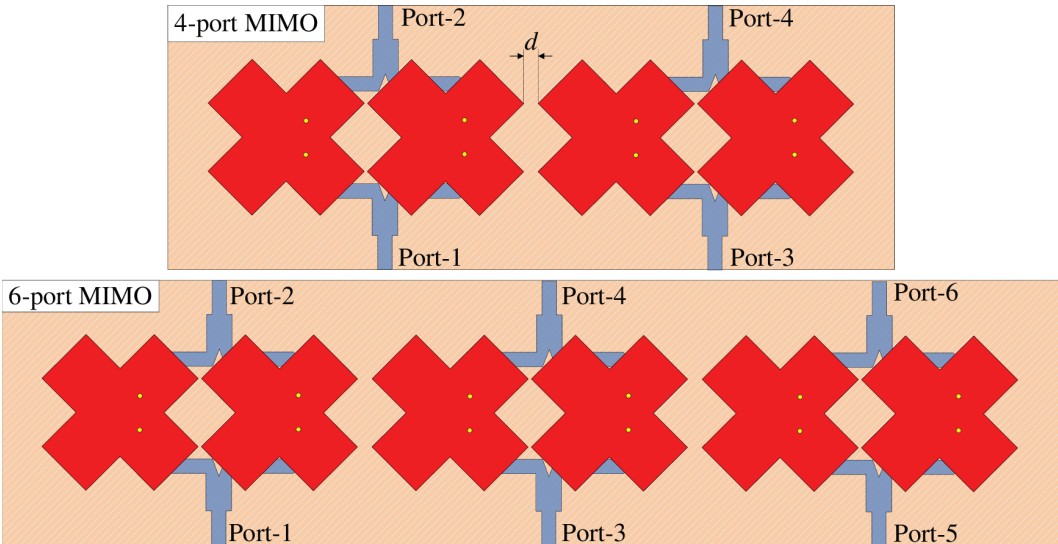

**Fig 11. Geometry of multi-port MIMO antennas.**

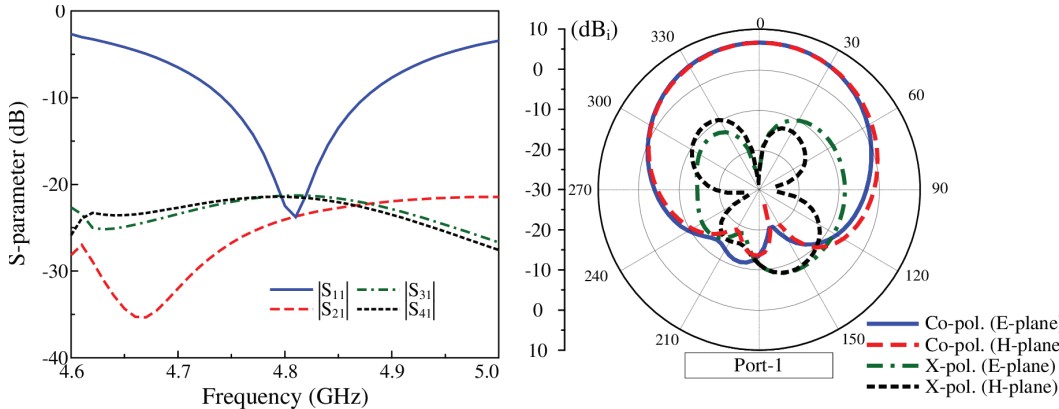

**Fig 12. Simulated S-parameter and radiation patterns at 4.8 GHz of the 4-port MIMO antenna.**

## Measurement results

To verify the design concept, a prototype of the 2-port MIMO antenna is fabricated and measured. The top- and bottom-view photos of the fabricated antenna are presented in Fig 15. Overall, the simulated and measured data are well matched with small discrepancy, which might be attributed to the tolerance of fabrication and the imperfection in measurement setup.

The simulated and measured S-parameter results are compared in Fig 16. The measured impedance bandwidth is 2.7% (4.74–4.87 GHz), which is quite close to the simulated data of 2.9% (4.72 –4.86 GHz). Meanwhile, the measured isolation across the impedance bandwidth of the fabricated antenna is always higher than 20 dB. There is small degradation with the simulation, which is better than 22 dB.

Fig 17 plots the radiation patterns at 4.8 GHz in two principal planes of E- and H-plane. Due to the symmetrical geometry between Port-1 and Port-2, only the measured data

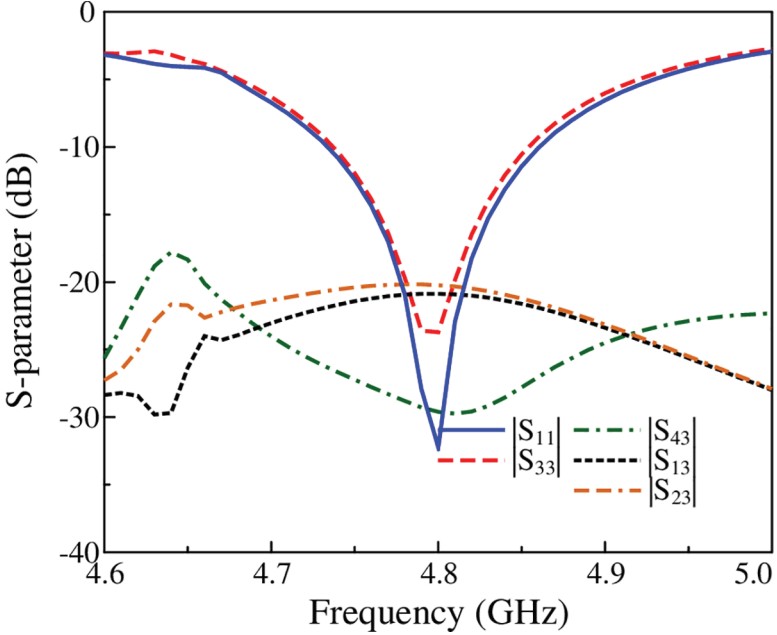

**Fig 13. Simulated S-parameter of the 6-port MIMO antenna.**

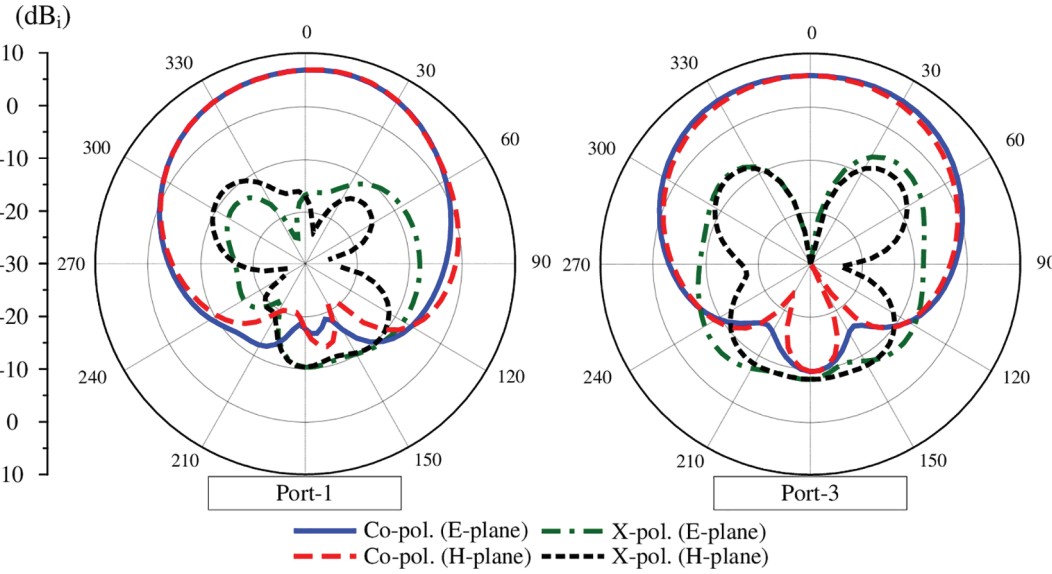

**Fig 14. Simulated gain radiation patterns at 4.8 GHz of the 6-port MIMO antenna.**

for Port-1 is presented. The data shows good radiation patterns in both planes, which are symmetrical around the broadside direction. The gain levels of the main lobe and back lobe are respectively 7.3 dBi and -10.8 dBi, leading to high front-to-back ratio (FBR) of 18.1 dB. Meanwhile, the cross-polarization is about 20 dB lower than the co-polarization.

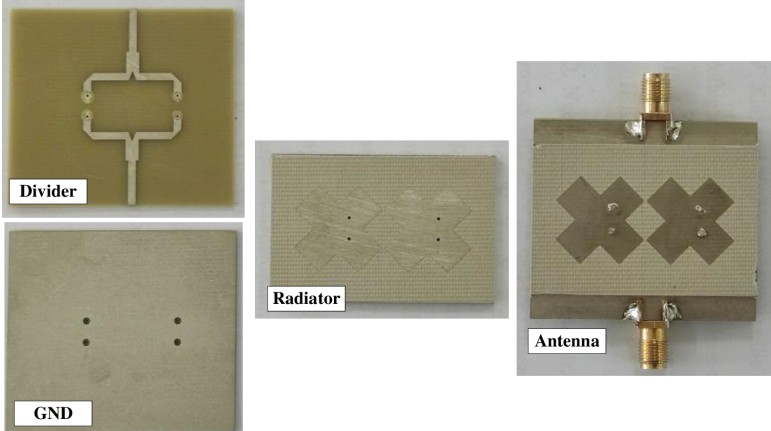

**Fig 15. Photographs of the fabricated 2-port MIMO antenna.**

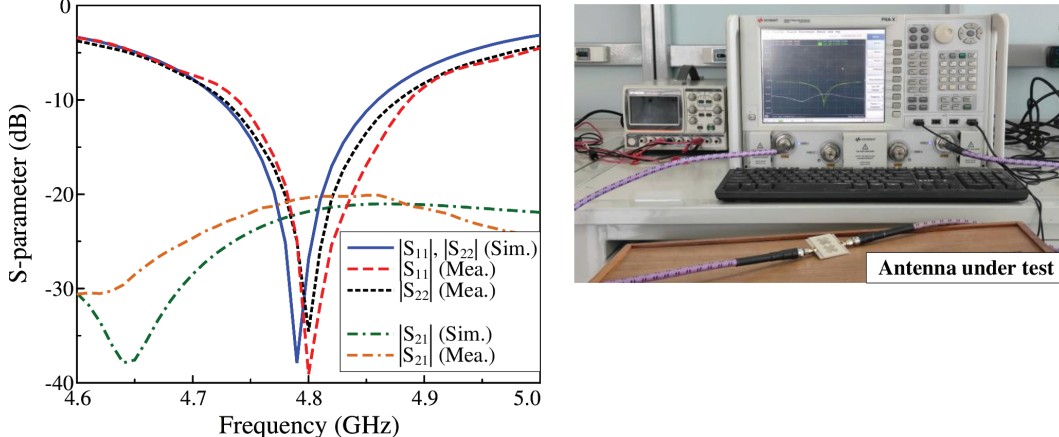

**Fig 16. Simulated and measured the S-parameter of the proposed 2-port MIMO antenna.**

## Performance comparison

Further demonstration of the advantages of the proposed approach over the other related works can be summarized in Table 2. Noted that all designs are compared with 2-port operation. It can be seen obviously that the proposed approach has the most compact size in terms of overall dimensions and element spacing as well. Besides, the use of dual-polarization patches can help to significantly reduce the number of radiating elements. High gain radiation can be achieved in [21,22,24] with the aid of FSS layers and large number of radiating elements, but they have a trade-off with antenna profiles and lateral dimensions. Meanwhile, the designs in [26,28] can perform higher gain, but larger number of radiating elements and larger size as well. It is also worth noting that the proposed approach is the first case utilizing dual-polarized patches to design high gain MIMO antenna.

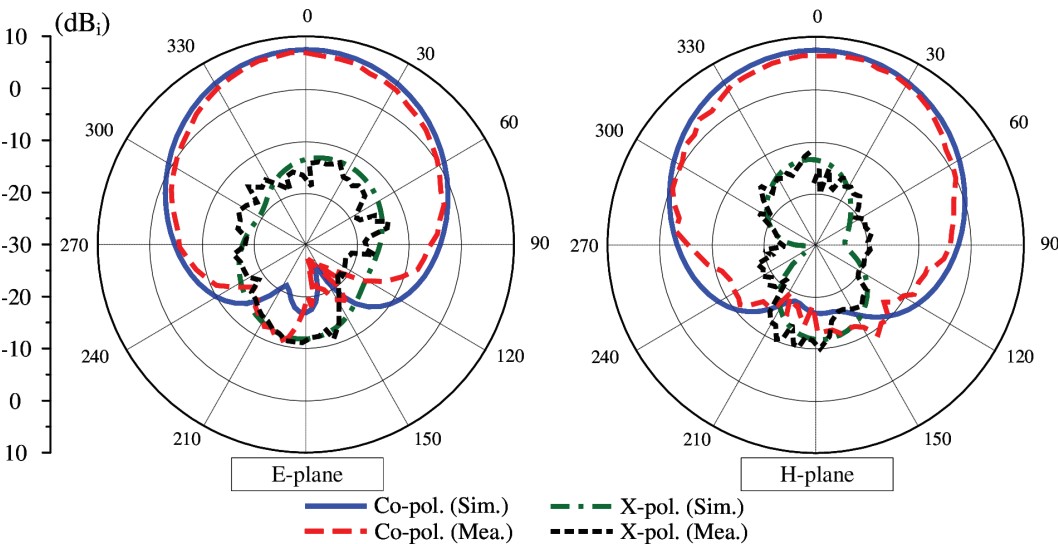

**Fig 17. Simulated and measured radiation patterns of the proposed 2-port MIMO antenna.**

**Table 2. Performance comparison among 2-port high gain MIMO antennas.**

| Ref. | Overall size ($\lambda^3$) | Polarization of radiator | No. of Elements | Spacing ($\lambda$) | Iso. (dB) | Gain (dBi) |
|------|---------|---------|---------|---------|---------|---------|
| [21] | $2.80 \times 1.40 \times 0.10$ | Single-pol. | 8 | 0.42 | 30 | 11.5 |
| [22] | $3.66 \times 2.55 \times 0.60$ | Single-pol. | 4 | 0.53 | 35 | 10.3 |
| [24] | $2.14 \times 1.16 \times 0.16$ | Single-pol. | 8 | 0.32 | 40 | 14.1 |
| [26] | $1.83 \times 1.83 \times 0.60$ | Single-pol. | 2 | 0.13 | 10 | 8.8 |
| [28] | $3.27 \times 2.81 \times 0.07$ | Single-pol. | 4 | 0.39 | 35 | 12 |
| [30] | $2.24 \times 2.24 \times 0.07$ | Single-pol. | 4 | 0.49 | 26 | 13 |
| Prop. | $0.72 \times 0.48 \times 0.04$ | Dual-pol. | 2 | 0.005 | 20 | 7.3 |

## Conclusion

This paper has proposed a method to design the MIMO antenna with compact size and high gain radiation. The proposed approach is a combination of dual-polarized crossed patches and T-junction power dividers. The 2-port MIMO antenna is fabricated and measured. This design with a compact size of $0.72\lambda \times 0.48\lambda \times 0.04\lambda$ has measured operating bandwidth of 2.7% and isolation of better than 20 dB. Meanwhile, the far-field results obtain high gain of 7.3 dBi and symmetrical radiation pattern around the broadside direction. Further investigation also demonstrates the capability of the proposed approach in multi-port MIMO configurations.

## Author contributions

**Investigation:** Phuong Kim-Thi.

**Project administration:** Tung Bui-Thanh.

**Supervision:** Tung Bui-Thanh.

**Writing – original draft:** Phuong Kim-Thi.

**Writing – review & editing:** Thang Nguyen-Van, Dat Nguyen-Tien.

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
