## [Decision Letter · Decision Letter 0]

17 Apr 2025

PONE-D-25-15830A Compact Design of MIMO Patch Antenna with High Gain and Symmetrical Radiation PatternPLOS ONE

Dear Dr. Kim,

Thank you for submitting your manuscript to PLOS ONE. After careful consideration, we feel that it has merit but does not fully meet PLOS ONE’s publication criteria as it currently stands. Therefore, we invite you to submit a revised version of the manuscript that addresses the points raised during the review process.

We look forward to receiving your revised manuscript.

Kind regards,

Sachin Kumar, Ph.D.

Academic Editor

PLOS ONE

Additional Editor Comments:

The authors must carefully respond to the reviewers' comments and submit the revised manuscript for further consideration.

Reviewers' comments:

Reviewer's Responses to Questions

**Comments to the Author**

1. Is the manuscript technically sound, and do the data support the conclusions?

Reviewer #1: Partly

Reviewer #2: Yes

2. Has the statistical analysis been performed appropriately and rigorously? 

Reviewer #1: Yes

Reviewer #2: Yes

3. Have the authors made all data underlying the findings in their manuscript fully available?

Reviewer #1: Yes

Reviewer #2: Yes

4. Is the manuscript presented in an intelligible fashion and written in standard English?

Reviewer #1: Yes

Reviewer #2: Yes

5. Review Comments to the Author

Reviewer #1: In this paper, the authors have presented the design of a compact multiple-input multiple-output antenna. The reflection coefficient of the antenna confirms the suitability of the proposed antenna for sub-6 GHz multiple-input multiple-output wireless systems. Following are the suggestions to the authors:

1. In the abstract, the abbreviation MIMO should be defined. All the abbreviations used first time in the paper should be defined.

2. Section numbering is missing.

3. In the introduction, some more recent literature related to sub-6GHz multiple-input multiple-output antennas should be included.

4. At the end of the introduction section, the structure of the paper should be given.

5. The dimensions of the antenna should be summarized in a table.

6. In Table 1, the unit of the dimension should be corrected as λ^3.

7. Appropriate tense should be used in conclusion, please rephrase first sentence of this section.

8. If not a journal requirement, please included figures within the main paper.

9. Include some diversity parameters such as diversity gain, envelope correlation coefficient etc.

10. As mentioned above, please include some more recent papers related to the sub-6 GHz multiple-input multiple-output antennas such as ‘Compact Sub 6 GHz Dual Band Twelve-Element MIMO Antenna for 5G Metal-Rimmed Smartphone Applications’, ‘Metasurface Superstrate based MIMO Patch Antennas with Reduced Mutual Coupling for 5G Communications’, ‘A closely spaced two-port MIMO antenna with a radiation null for out-of-band suppressions for 5G Sub-6 GHz applications’ etc.

11. The details of the reference [26] are incomplete.

Reviewer #2: 1. The authors have presented a good research analysis, for the power divider network provide some surface current analysis.

2. How isolation is achieved must be discussed briefly.

3. Provide some information on the radiation pattern with back lobe suppression characteristics.

4. Highlight some novelty of the proposed design.

5. For the array antenna design how did the authors reduce side lobes.

6. PLOS authors have the option to publish the peer review history of their article (what does this mean?). If published, this will include your full peer review and any attached files.

Reviewer #1: No

Reviewer #2: **Yes: **Asutosh mohanty

---

## [Author Response · Author response to Decision Letter 1]

25 Apr 2025

Submission ID: PONE-D-25-15830

Original Article Title: “A Compact Design of MIMO Patch Antenna with High Gain and Symmetrical Radiation Pattern”

To: Reviewer

Re: Response to reviewer

Dear Reviewer,

We appreciate you for your precious time in reviewing our paper and providing valuable comments. It was your valuable and insightful comments that led to possible improvements in the current version. The authors have carefully considered the comments and tried our best to address every one of them.

We are uploading our point-by-point response to the comments, an updated manuscript with red highlighting indicating changes, and a manuscript without track changes.

Best regards,

Reviewer 1: In this paper, the authors have presented the design of a compact multiple-input multiple-output antenna. The reflection coefficient of the antenna confirms the suitability of the proposed antenna for sub-6 GHz multiple-input multiple-output wireless systems. Following are the suggestions to the authors:

Concern # 1: In the abstract, the abbreviation MIMO should be defined. All the abbreviations used first time in the paper should be defined.

Author response: Agreed.

Author action: The abbreviation MIMO is defined in Abstract of the revised manuscript.

Concern # 2: Section numbering is missing.

Author response: The Section number is missing due to the template of the PLoS One journal.

Concern # 3: In the introduction, some more recent literature related to sub-6GHz multiple-input multiple-output antennas should be included.

Author response: Agreed.

Author action: Several recent published papers related to sub-6 GHz MIMO antenna are added to the revised manuscript as ref [9, 10, 15].

Concern # 4: At the end of the introduction section, the structure of the paper should be given.

Author response: Agreed.

Author action: The structure of the paper is added at the end of the Introduction Section in the revised manuscript.

Concern # 5: The dimensions of the antenna should be summarized in a table.

Author response: Agreed.

Author action: The dimensions of the antenna are summarized in Table 1 of the revised manuscript.

Concern # 6: In Table 1, the unit of the dimension should be corrected as λ^3.

Author response: Agreed.

Author action: The unit of the dimension is corrected in the revised manuscript.

Concern # 7: Appropriate tense should be used in conclusion, please rephrase first sentence of this section.

Author response: Agreed.

Author action: The first sentence in the Conclusion is rephrased in the revised manuscript.

Concern # 8: If not a journal requirement, please included figures within the main paper.

Author response: The authors agree with the Reviewer that including figures within the main paper is better. However, according to the journal template, it is not allowed in this stage.

Concern # 9: Include some diversity parameters such as diversity gain, envelope correlation coefficient etc.

Author response: Agreed.

Author action: The ECC and DG of the 2-port MIMO antenna have been included in the revised manuscript as Fig. 10.

Concern # 10: As mentioned above, please include some more recent papers related to the sub-6 GHz multiple-input multiple-output antennas such as ‘Compact Sub 6 GHz Dual Band Twelve-Element MIMO Antenna for 5G Metal-Rimmed Smartphone Applications’, ‘Metasurface Superstrate based MIMO Patch Antennas with Reduced Mutual Coupling for 5G Communications’, ‘A closely spaced two-port MIMO antenna with a radiation null for out-of-band suppressions for 5G Sub-6 GHz applications’ etc.

Author response: The authors would like to thank the Reviewer for the valuable suggested references.

Author action: The suggested references are included in the revised manuscript as refs [9, 10, 15].

Concern # 11: The details of the reference [26] are incomplete.

Author response: The authors would like to thank the Reviewer for pointing out the authors’ mistake.

Author action: The details of ref [26] are corrected in the revised manuscript.

Reviewer 2:

Concern # 1: The authors have presented a good research analysis, for the power divider network provide some surface current analysis.

Author response: In accordance with the Reviewer comment, the surface current on the T-divider is shown in Fig. 1R. As shown, the current is equally distributed to the outputs of the divider. This makes sure that the radiating elements are excited with equal magnitude and phase.

This is the common operation of the T-divider and thus, the authors do not include this figure in the manuscript for brevity.

Fig. 1R. Simulated current distribution at 4.8 GHz on the divider.

Concern # 2: How isolation is achieved must be discussed briefly.

Author response: The authors would like to thank the Reviewer for the constructive comment. The principle for high isolation of the proposed design is based on two orthogonal linear polarizations. As seen in Fig. 5, F-1 and F-3 will be excited for Port-1 operation. The coupling will occur between these feeding positions to the others, F-2 and F-4. However, the positions of F-2 and F-4 are orthogonal to F-1 and F-3. Consequently, high isolation will be obtained, as demonstrated in Section “Design a dual-polarized antenna”.

Author action: A brief discussion about the isolation of 2-port MIMO is added to Paragraph 1, Subsection “2-port MIMO”, Section “MIMO configurations” of the revised manuscript.

Concern # 3: Provide some information on the radiation pattern with back lobe suppression characteristics.

Author response: Agreed.

Author action: Further discussion is added to Paragraph 3, Subsection “2-port MIMO”, Section “MIMO configurations” of the revised manuscript.

Concern # 4: Highlight some novelty of the proposed design.

Author response: Agreed.

Author action: The novelty of the proposed design is emphasized in Paragraph 4, Section “Introduction” of the revised manuscript.

Concern # 5: For the array antenna design how did the authors reduce side lobes.

Author response: For the MIMO array, the side lobes can be reduced by controlling excitation magnitudes and/or phases. This task requires complicated feeding networks as they distribute power and phase to each antenna element independently. In this paper, the authors just presented a MIMO antenna with high isolation and high gain operation, rather than proposing the feeding network.

---

## [Decision Letter · Decision Letter 1]

30 Apr 2025

PONE-D-25-15830R1A Compact Design of MIMO Patch Antenna with High Gain and Symmetrical Radiation PatternPLOS ONE

Dear Dr. Kim,

Thank you for submitting your manuscript to PLOS ONE. After careful consideration, we feel that it has merit but does not fully meet PLOS ONE’s publication criteria as it currently stands. Therefore, we invite you to submit a revised version of the manuscript that addresses the points raised during the review process.

We look forward to receiving your revised manuscript.

Kind regards,

Sachin Kumar, Ph.D.

Academic Editor

PLOS ONE

Journal Requirements:

Additional Editor Comments:

The reviewers are requested to change the 'S-parameter' to 'Diversity Gain' on the right vertical axis in Fig. 10.

Reviewers' comments:

Reviewer's Responses to Questions

**Comments to the Author**

1. If the authors have adequately addressed your comments raised in a previous round of review and you feel that this manuscript is now acceptable for publication, you may indicate that here to bypass the “Comments to the Author” section, enter your conflict of interest statement in the “Confidential to Editor” section, and submit your "Accept" recommendation.

Reviewer #1: All comments have been addressed

Reviewer #2: All comments have been addressed

2. Is the manuscript technically sound, and do the data support the conclusions?

Reviewer #1: Yes

Reviewer #2: Yes

3. Has the statistical analysis been performed appropriately and rigorously? 

Reviewer #1: Yes

Reviewer #2: N/A

4. Have the authors made all data underlying the findings in their manuscript fully available?

Reviewer #1: Yes

Reviewer #2: Yes

5. Is the manuscript presented in an intelligible fashion and written in standard English?

Reviewer #1: Yes

Reviewer #2: Yes

6. Review Comments to the Author

Reviewer #1: Authors have addressed the comments in the revised paper. In the Fig 10 of the revised manuscript, please correct ‘S-parameter’ as ‘Diversity Gain’ on the right vertical axis.

Reviewer #2: (No Response)

7. PLOS authors have the option to publish the peer review history of their article (what does this mean?). If published, this will include your full peer review and any attached files.

Reviewer #1: No

Reviewer #2: **Yes: **Asutosh mohanty

---

## [Author Response · Author response to Decision Letter 2]

1 May 2025

Submission ID: PONE-D-15830R1

Original Article Title: “A Compact Design of MIMO Patch Antenna with High Gain and Symmetrical Radiation Pattern”

To: Reviewer

Re: Response to reviewer

Dear Reviewer,

We appreciate you for your precious time in reviewing our paper and providing valuable comments. It was your valuable and insightful comments that led to possible improvements in the current version. The authors have carefully considered the comments and tried our best to address every one of them.

We are uploading our point-by-point response to the comments, an updated manuscript with red highlighting indicating changes, and a manuscript without track changes.

Best regards,

Reviewer 1:

Concern # 1: Authors have addressed the comments in the revised paper. In Fig 10 of the revised manuscript, please correct ‘S-parameter’ as ‘Diversity Gain’ on the right vertical axis.

Author response: The authors would like to thank the Reviewer for pointing out our mistake.

Author action: Fig. 10 is corrected in the revised manuscript.

Journal Requirements: Please review your reference list to ensure that it is complete and correct. If you have cited papers that have been retracted, please include the rationale for doing so in the manuscript text, or remove these references and replace them with relevant current references. Any changes to the reference list should be mentioned in the rebuttal letter that accompanies your revised manuscript. If you need to cite a retracted article, indicate the article’s retracted status in the References list and also include a citation and full reference for the retraction notice.

Author response: The authors have thoroughly checked the references. All references are completed and corrected. No retracted references are cited in the paper.

---

## [Editor Report · Decision Letter 2]

12 May 2025

A Compact Design of MIMO Patch Antenna with High Gain and Symmetrical Radiation Pattern

PONE-D-25-15830R2

Dear Dr. Kim,

We’re pleased to inform you that your manuscript has been judged scientifically suitable for publication and will be formally accepted for publication once it meets all outstanding technical requirements.

Kind regards,

Sachin Kumar, Ph.D.

Academic Editor

PLOS ONE
---

## [Editor Report · Acceptance letter]

PONE-D-25-15830R2

PLOS ONE

Dear Dr. Kim-Thi,

I'm pleased to inform you that your manuscript has been deemed suitable for publication in PLOS ONE. Congratulations! Your manuscript is now being handed over to our production team.

Kind regards,

on behalf of

Dr. Sachin Kumar

Academic Editor

PLOS ONE